# OpenReview forum: "Sparks of Cooperative Reasoning: LLMs as Strategic Hanabi Agents"
_ICLR.cc/2026/Conference — Submitted to ICLR 2026_

### Official Review · Reviewer_nK8E · 2025-10-24

**Soundness:** 3
**Presentation:** 3
**Contribution:** 3
**Rating:** 4
**Confidence:** 3

**Summary:**

This paper investigates the use of Hanabi as a benchmark for evaluating large language models (LLMs) in cooperative reasoning. The authors adapt the game into a multi-turn, multi-agent format and evaluate a diverse set of LLMs under three prompting schemes (Watson, Sherlock, and Mycroft) which vary the degree of explicit and implicit information accessible to each agent. Beyond evaluation, the paper introduces two new Hanabi datasets. The authors further show that a smaller model fine-tuned (SFT) on this new data substantially outperforms strong baselines and existing state-of-the-art models on their cooperative reasoning benchmark. Overall, the paper positions Hanabi as a controlled yet expressive environment for testing an LLM’s ability to deduce hidden information from evolving states and coordinate with others based on incomplete knowledge.

**Strengths:**

- Extensive model coverage: The paper evaluates a wide range of LLMs under three distinct prompting setups (Watson, Sherlock, Mycroft) and multiple random seeds, providing a thorough view of model performance across configurations.
- Benchmark insight: The work effectively demonstrates that Hanabi, when reformulated under their multi-agent prompting scheme, can serve as a useful benchmark for studying implicit reasoning and cooperative inference in LLMs.
- Transparency and reproducibility: The experimental setup, prompts, and evaluation details are well-documented, ensuring replicability.
- Clear motivation: The paper makes a compelling case for moving beyond single-agent reasoning benchmarks toward cooperative, belief-dependent tasks.

**Weaknesses:**

- The novelty relative to prior benchmark studies (like those mentioned LLM-Arena and SPIN-Bench) is not discussed in depth enough beyond their introduction in the related works sections.
- The generalisation value of fine-tuning a specialised Hanabi model remains limited. It is unclear whether such training improves reasoning in other cooperative or belief-based tasks, and thus the benefit of both the datasets and the SFT results remain unclear.

**Questions:**

1. Could you clarify how your approach differs from a “single prompt per turn” setup, as stated in the related works section.
2. You mention “adding Hanabi strategies.” Could you elaborate on what these strategies are, and how they are represented or introduced into the model’s reasoning process?
3. It would be helpful to understand why non-reasoning models performed worse with added contextual information. Does this suggest overfitting to irrelevant details or difficulty managing longer prompts?
4. You note that there are exceptions to the performance vs number of agents pattern. performance. Do you have any insight into why these exceptions occur? Might it have to do with the unique strategies each agent uses?
5. Regarding Mycroft: why not rely on the model’s own deductions to update its knowledge about its hand?
    - Related: If a model fails to deduce something correctly or omits key information, is that information permanently lost from its belief state, or can it be recovered later.

---

> ### Author Response · Authors · 2025-11-25
>
> We thank the reviewer for their valuable feedback, and glad the reviewer found our work **clearly motivated** and **useful to study implicit reasoning in LLMs**, and that they highlight the **coverage and transparency of our evaluation**. We particularly appreciate the effort the reviewer put into the feedback and questions, which helped us to improve the paper.
>
> ### Weaknesses
> ---
> **W1 and Q1.**  A key limitation of both LLM‑Arena and SPIN‑Bench is that they delegate most of the hard Hanabi reasoning to the game engine: on every turn, the agent is given a fresh snapshot of the game plus programmatic deductions about what each card “could be,” so the environment, not the model, is effectively doing the long‑horizon state tracking (We mentioned this in “limitations of Sherlock”). In contrast, our main novelty is to remove engine‑provided deductions and force the model to carry its own working memory across turns:
> 1. In Mycroft (Sec. 5), we no longer supply “could be” lists or game‑engine history. Each agent receives only the public state plus its own previous deduction block and last action and must update that block itself, mimicking how humans track possibilities over time.
> 2. To give more clarity we added an LLM‑judge evaluation to the revision (Appendix E.3) that scores models on Deduction Accuracy, History Integration, and State‑Tracking Quality, directly measuring abilities that prior benchmarks leave to the engine. Our analysis shows that even advanced reasoning models still have significant room to improve at state-tracking (Table 3).
> 3. We also provide the input and output prompts of all three setups (Appendix I.1, I.2 and I.6).
> We believe this kind of explicit, agent‑side state tracking is increasingly important for modern long‑horizon agents (e.g., code assistants and tools‑using “coding agents”) that must keep a consistent working memory over many steps.
> W2. We thank the reviewer for highlighting this point. To show the potential of our dataset, we train Qwen3-4B on our HanabiRewards dataset with RL and observe a 156% improvement to score (+7.5) in contrast to our previous 21% (+1.0) improvement with only instruction tuning (Fig. 7, Section 6.2 in the revision). As we show in Section 6.3, training on HanabiRewards also improves the performance in out-of-domain cooperative reasoning (Guessing Game), Temporal reasoning (EventQA), Instruction following (IFBench) and doesn’t degrade general reasoning capability of the pretrained model (AIME 2025).
> ### Questions
> ---
> **Q2.** In Sherlock (DeductCon) and in the MoA aggregator prompts, we encode Hanabi strategies as natural-language rubrics that guide decision-making. We detail our exact prompts for strategies in Appendix I.2, specifically "Evaluate Playable Cards in Hand’’, "Consider Teammates’ Hands and Hint Opportunities’’ and "Assess Discard Options to Gain Info Tokens.’’
> In contrast to prior work, we give the agent the initial card distribution and ask the agent to calculate the probability of each card in its hand before taking actions (see App I.2 for our exact prompt). In addition, we also give general Hanabi playing strategies like "Try to save the critical cards like rank 5, second card of each color, rank 2, 3, 4.’’
>
> **Q3.**  Our analysis suggests the issue is not long prompts, but the inability of non-reasoning models to weigh different instructions. We discuss this in more detail in Appendix C and added a qualitative example in Appendix J.1. We point to this in the main paper revision in Sec. 4 (L317).
>
> **Q4.** With respect to exceptions to Hanabi performance having an inverse relationship to number of players, we add the following details for more insight:
> In the Mycroft setting, as we increase from 3→5 players, average scores decrease (Fig 17, App D.2) which is expected as each extra player adds more hands, more hints, and a more complex belief state for the model to maintain. As an exception, 2-player performance is lower than 3-player on all models except o3. This happens because when there are only two players, any mistake from inconsistent state tracking is more severe. We also observe similar difficulty in state tracking with increasing player count with our RL finetuned Qwen3-4B model, which we report in Appendix H.2 (Fig. 29).
> The remaining exceptions appear to be strategy‑driven. In Appendix C we describe several emergent behaviors: for example, Gemini 2.5 Pro plays aggressively until it has lost two life tokens and then switches to conservative play. These kinds of idiosyncratic conventions and risk profiles can make a particular player configuration (Mostly 2-player) look slightly better or worse than the surrounding ones, even though the overall trend with more players is downward.

---

> > ### Author Response · Authors · 2025-11-25
> >
> > ### Continued
> > ---
> > **Q5.** We thank the reviewer for this pointer. In fact our Mycroft setting *does* rely on its own deductions, we inadvertently had an older prompt variation for the Mycroft setup in the paper; we have now corrected it.
> > *“If a model fails to deduce something correctly or omits key information, is that information permanently lost from its belief state, or can it be recovered later?”*
> > If the agent forgets to incorporate a piece of negative information (e.g., “this card cannot be green”), that omission is not recoverable in the future. This design is intentional: it mirrors human play where missed deductions or mis-remembered information can permanently handicap a player’s belief state. We use an LLM as a judge to quantify how often such errors occur, and observe a large performance drop from Sherlock to Mycroft in Appendix E.3.
> >
> > We hope that the rebuttal clarifies the concerns and questions raised by the reviewer. We would be **very happy to discuss any further questions about the work**, and would really appreciate an appropriate increase in the score if the reviewer’s concerns are adequately addressed to facilitate acceptance of the paper.

---

> > > ### Comment · Reviewer_nK8E · 2025-11-25
> > >
> > > I would like to thank the authors for their thoughtful and thorough response. You have effectively answered all of my questions and, most importantly, addressed my primary concern regarding generalisability. Consequently, I will adjust my score accordingly.

---

> > > > ### Author Response · Authors · 2025-11-25
> > > >
> > > > We thank the reviewer for their prompt response and really appreciate the adjustment in score reflecting that we addressed their questions and their primary concern regarding generalization. We sincerely thank the reviewer for their time and effort in the review process and helping us improve the quality of our work.

---

### Official Review · Reviewer_aPy3 · 2025-10-29

**Soundness:** 3
**Presentation:** 2
**Contribution:** 3
**Rating:** 8
**Confidence:** 3

**Summary:**

This paper presents a large-scale evaluation of 17 Large Language Models (LLMs) on their ability to perform cooperative reasoning in the challenging incomplete-information card game, Hanabi. The authors investigate performance across 2-5 player settings and explore the impact of different context scaffolding techniques: MinCon (minimal state), DeductCon (state plus explicit deductions from the game engine), and Mycroft (a multi-turn setup requiring implicit state tracking by the LLM) . Key findings indicate that reasoning-enhanced LLMs outperform non-reasoning ones, and explicit deductions (DeductCon) significantly boost performance (average scores >15/25), although still falling short of expert humans and specialized Hanabi agents (>20/25). The Mycroft setting proved difficult due to state-tracking challenges. The paper also contributes two valuable datasets: HanabiLogs (1,520 game trajectories) and HanabiRewards (560 games with dense move-level value annotations). Instruction-tuning a small LLM (Qwen-3-4B) on HanabiLogs demonstrated a 21% score improvement, surpassing some larger closed models in the DeductCon setting.

**Strengths:**

**Scale and Scope:** The evaluation is comprehensive, covering a wide range of recent LLMs (17 models), multiple player counts (2-5), and includes cross-play experiments. This provides a robust snapshot of current LLM capabilities in Hanabi.

**Dataset Contributions:** HanabiLogs and HanabiRewards are significant contributions, providing richly annotated data with move utilities, which can fuel future research in instruction tuning and reinforcement learning for cooperative agents. The successful SFT demonstration validates HanabiLogs' utility.


**Systematic Scaffolding Analysis:** The comparison between MinCon, DeductCon, and Mycroft offers clear insights into how context, explicit knowledge, and implicit state tracking affect LLM performance in cooperative reasoning tasks.


**Transparency and Positioning:** The paper clearly articulates its methodology, acknowledges limitations of prior work regarding reproducibility , and provides context relative to existing benchmarks and agents .

**Weaknesses:**

**Scaffolding Choices:** While insightful, the performance heavily depends on the chosen scaffolding (MinCon, DeductCon). It remains unclear why these specific strategies are chosen, and makes readers wonder about generalization to settings without such explicit support, different context, unseen partners and open-ended conventions.

**Implicit State Tracking Analysis:** The Mycroft setting highlights difficulties with implicit state tracking, but the analysis doesn't quantify the source of error (e.g., context limits vs. reasoning failure) or evaluate the quality of the state/deductions the models attempted to track.

**Inconsistencies:** In the abstract on OpenReview, the improvement via instruction tuning on HanabiLogs is 24%, whereas it becomes 21% in the abstract of the pdf version paper.

**Questions:**

1. Could you provide a more detailed ablation study isolating the contribution of each component within the Sherlock (DeductCon) prompt (e.g., strategic advice vs. explicit deductions)? Are the full prompts used for Watson, Sherlock, and Mycroft available?

2. Could you report contamination checks and take the fact that models may be exposed to Hanabi tutorials/solutions into consideration?

3. Have you considered or conducted experiments involving human-LLM cross-play, particularly with human players unfamiliar with typical LLM strategies, to assess adaptability and convention formation?

4. How were the dense move utilities in HanabiRewards generated? Do they rely on heuristics, model estimates, search algorithms, or human annotations? Clarity on their origin is important for their use in RL.

---

> ### Author Response · Authors · 2025-11-25
>
> We thank the reviewer for their positive feedback and insightful questions. We are glad the reviewer found our datasets to be **significant contributions** that can **fuel future research in cooperative agents** and they found our evaluation **systematic, comprehensive and transparent**. Below we address comments from the review:
>
> ### Weaknesses
> ---
>
> **Scaffolding Choices**. We use 3 different prompts with increasing complexity and gradually removing explicit information to test how models reason on their own and how they can rank the choices that are deduced based on the information provided to them. Watson (MinCon) establishes a baseline by testing whether LLMs possess inherent knowledge of Hanabi rules and basic cooperative strategies without external guidance. This evaluates "out-of-the-box" cooperative reasoning capabilities. Sherlock (DeductCon) adds explicit game engine deductions and strategic advice to determine whether LLMs can leverage rich contextual information when provided. Lastly, Mycroft removes explicit game engine deductions to evaluate whether LLMs can autonomously track and update belief states across multiple turns.
> Beyond scaffolding, we also evaluate generalization:
> In our original submission, we evaluate LLM cross-play (Section 6.1) to evaluate cooperation between agents with a large skill gap (Grok-3-mini + o4-mini), showing cross-play performance interpolates between self-play baselines.
> In the revision, we observe that training on HanabiRewards transfers to other cooperative and temporal reasoning tasks (Group Guessing Game, EventQA) and instruction following (IFBench) while not degrading reasoning learned during pretraining (AIME). We provide details in Section 6.3
>
> **Implicit State Tracking Analysis**. We thank the reviewer for highlighting this point, we have added a qualitative example in Appendix J.1 to analyze this behavior. We observe that the model's low scoring in the Mycroft setup is due to its failure to make deductions and track states. Context limits were not a problem, as seen from the example (Appendix I.6). We also use an LLM judge to compare implicit model state tracking with gold deductions in Appendix E.3, and we show that even advanced reasoning models still have significant room to improve at state-tracking (Table 3).
> **Inconsistencies**. We have fixed this in the revision and on OpenReview, and thank the reviewer for pointing this out.
>
> ### Questions
> ---
>
> **Q1**. We discuss the impact of the various components of Sherlock in Section 3.2, where we systematically compare: MinCon (Watson) with minimal context, SPIN-Bench, SPIN-Bench without engine deductions, SPIN-Bench without deductions and discard pile, and our Sherlock setting that adds probability-based reasoning + strategies. In the revision, we provide exact prompts in Appendix. (I.1 - Watson, I.2 -Sherlock, I.6 - Mycroft)
>
> **Q2**. In general, we find that models *do know about Hanabi*. We indirectly measure this contamination through model scores in the Watson setting, since we do not provide any Hanabi strategies to the agent. We also highlighted this in our original submission in Appendix C, where we discuss how Mistral models perform well in the Sherlock setting compared to Watson due to limited pretrained knowledge about Hanabi. We would also like to point out that knowledge of Hanabi rules is not true contamination, since to do well at the game the models still have to deduce information from the evolving game state and other player actions. We view this as analogous to the standard RL game setting that assumes human players know game rules: the benchmark then probes how models reason given those rules, not whether they memorized them. While evaluating fine-tuned Qwen3-4B (HanabiLogs/HanabiRewards), we use disjoint seeds from those used to collect trajectories to mitigate memorization (Appendix B).
>
> **Q3**. We are also intrigued by this experiment. In this work, we chose to focus on fully reproducible, scalable LLM-LLM settings for two reasons:
> Running human-LLM experiments to the same scale as we did with LLM-LLM, i.e. across 17 models, 4 player counts, and multiple prompts, would require substantial annotation overhead.
> Our goal was to first establish *where LLM agents break down in controlled settings* (Watson/Sherlock/Mycroft, cross-play) and to release datasets that enable more systematic training of models for cooperation. We briefly discuss human-LLM collaboration in Sec. 7, referencing AH2AC2 (Dizdarevic et al., 2024), as a key avenue for future research. We hope that this work serves as a foundational positive step in this direction.

---

> ### Author Response · Authors · 2025-11-25
>
> ### Continued
> ---
> **Q4**. The dense utilities in HanabiRewards are generated by LLM self-evaluation during game play. During each Watson/Sherlock/Mycroft game, we prompt an LLM to: Enumerate all legal actions, Provide a chain-of-thought justification, and Assign a scalar rating in [-1, 1] to each legal move before selecting its chosen action. We briefly mention this in Section 3.1 in the original submission. We agree that providing these details are important; we also add extensive details on all setups in Appendix I in the revision.
>
> We hope that the rebuttal clarifies the concerns and questions raised by the reviewer. We would be **very happy to discuss any further questions about the work** to further improve its quality to facilitate acceptance of the paper.

---

> > ### Comment · Reviewer_aPy3 · 2025-11-25
> > **Thank you for the response.**
> >
> > Thank you to the authors for their detailed response. The response makes sense to me, and I believe it's a good paper that warrants acceptance. I will maintain my score.

---

> > > ### Author Response · Authors · 2025-11-26
> > >
> > > We thank the reviewer for their response and efforts during the review cycle and are very glad they feel our work **warrants acceptance**. We sincerely thank the reviewer for their detailed feedback and questions which helped us revise and improve our work.

---

### Official Review · Reviewer_PfUB · 2025-11-01

**Soundness:** 2
**Presentation:** 2
**Contribution:** 2
**Rating:** 4
**Confidence:** 5

**Summary:**

The authors evaluate a suite of frontier language models on the Hanabi Learning Environment. The HLE is a popular framework in multi-agent reinforcement learning to evaluate agents' abilities to coordinate and cooperate. The authors reveal a significant discrepancy in performance between frontier models, although with seemingly high variance. They also show that additional information is not necessarily beneficial to performance for all models. However, for the models where it is beneficial, these models struggle to maintain a game history themselves, and performance declines, although it is unclear whether this is statistically significant.

**Strengths:**

1. The authors are transparent about the metrics they report and the hyperparameters they use.
2. The plots are mostly well-formatted and easy to read.
3. The paper follows a logical flow and is easy to follow.
4. The experiments seem reasonable.
5. Evaluating LLMs on coordination and cooperation tasks is highly significant for the community, and reporting trustworthy results on a popular benchmark is very useful. Furthermore, fine-tuning data can also be highly impactful and useful.

**Weaknesses:**

## Clarity
1. Citation formatting is inconsistent.
2. Sometimes citations are missing, like for BAD and SAD.
3. The writing tense changes

## Quality
1. Overall, while the authors provide the standard deviations, they do not seem to account for them when making conclusions. It appears that most standard deviations heavily overlap, and it's unclear whether any of the conclusions hold up under a statistical significance test. Especially with such high variance, the authors might want to consider using libraries like rliable (https://agarwl.github.io/rliable/) to make conclusions confidently. For example, the authors state

> Our high-level goal is to evaluate and improve the cooperative capabilities of LLMs in multi-agent settings, which we do in this work through the lens of Hanabi.

But none of the improvements via the fine-tuning dataset appear statistically significant.

## Soundness
1. The fine-tuning is motivated by an increase in performance and matching the performance of "strong" closed-source models like GPT-4o and Grok 3. However, these models rank among the lowest performing models, as shown in Figure 3 in DeductCon. The authors should consider adapting the language.

**Questions:**

1. Please consider increasing the size of the legend in Figure 4.
2. Would the authors consider doing a more thorough analysis if the performance rankings and changes in average performance hold up under any statistical significance test? Currently, it's extremely challenging to discuss experiments and propose potential improvements when most of them do not discriminate between models or game settings.

If the authors could address the above concerns, I'm looking forward to continuing the discussion during the rebuttal period.

---

> ### Author Response · Authors · 2025-11-25
>
> We thank the reviewer for their feedback and glad they found our work **highly significant for the community** and highlighted our dataset contributions to be **highly impactful and useful** . Below we address review comments:
>
> ### Weaknesses
> ---
>
> **Clarity**: We thank the reviewer for pointing out inconsistent and missing citations, we have fixed these in the revision. We have also made an effort to make tense consistent throughout the main paper.
>
> **Quality**: As per the reviewer's suggestion, we evaluate the inter-quartile mean (IQM) and 95% confidence interval scores for all our experimental results from the main paper using rliable library [1]. We discuss these results in Fig 10-16 (Appendix D1), and observe that they concur with the following trends from the main paper:
>
> 1. Figs 10-11, confirms the separation between reasoning and non-reasoning models, the ranking and qualitative trends from Section 4 remain unchanged.
> 2. Fig 12, confirms that when compared to the Sherlock setup, mycroft yields a significant drop for all reasoning models, thus verifying our claim that multi-turn state tracking is substantially harder.
> 3. The IQM with 95% confidence plot in Fig 13, confirms that Grok-3-mini + o4 mini cross play lies between the two self-play baselines as claimed in Sec 6.1
> 4. Apart from the significant tests on the basic evaluations, we also confirm our SFT + RLVR experiments on Qwen3-4B in the Sherlock setting. Although Fig. 14 shows a slight reduction in difference between the base Qwen3-4B and the HanabiLogs fine-tuned model (from ~1.0 to ~0.6), the ranking and direction of the change remain consistent.
>
> Lastly, we extend our prior instruction tuning on HanabiLogs to reinforcement learning with verifiable rewards on HanabiRewards with a Qwen3-4B backbone. We now observe a 7.5 point increase with the Sherlock setting (+156% over the base model) in contrast to the previously reported 1.0 point increase (+21%). We show in the main paper (Fig 7 and 8) and with rliable (Fig 15, Appendix D.1) that these results are statistically significant with variance and 95% Confidence Interval.
>
>
> **Soundness**: We thank the reviewer for pointing out the “strong” closed-source model claims for Grok-3 and GPT-4o although they were performing poorly. We have changed the wording in the revision to better reflect this (Sec 6.2). We now state that our goal is to show that a small open instruction model can, with our HanabiLogs/HanabiRewards dataset, surpass all non-reasoning proprietary models (beating the best performing GPT-4.1 by 4.2 points) and *narrow the remaining gap* to frontier reasoning models like o4‑mini to within 3 points.
>
>
>
> ### Questions
> ---
> **Q1**: We have increased the size of legends and font in all the diagrams to improve readability.
>
> **Q2**: As discussed in the earlier comment (**Quality**), we find consistent trends using a 95% confidence interval with rliable (Appendix D.1). We are more than happy to discuss any further experiments or potential improvements to further increase the quality and soundness of our work.
>
> We hope that the rebuttal clarifies the concerns and questions raised by the reviewer. We would really appreciate an appropriate increase in the score if the reviewer’s concerns are adequately addressed to facilitate acceptance of the paper, and in engaging in further discussion.
>
> ### References
> ---
>
> [1] Agarwal et al. Deep reinforcement learning at the edge of the statistical precipice. NeurIPS, 2021.

---

### Official Review · Reviewer_EE5A · 2025-11-03

**Soundness:** 2
**Presentation:** 3
**Contribution:** 2
**Rating:** 2
**Confidence:** 3

**Summary:**

This paper evaluates the cooperative reasoning abilities of 17 state-of-the-art LLMs in the challenging multi-agent card game Hanabi. By designing and investigating various prompting strategies, deductive context with engine-provided inferences (Sherlock/DeductCon), and a novel multi-turn implicit deduction regime (Mycroft). The authors aim to dissect the limitations and capabilities of LLMs in strategic, theory-of-mind based cooperation under incomplete information. The study includes extensive empirical benchmarking across 2-5 player settings, a broad cross-section of LLMs, ablation studies on context engineering and team composition, and the creation of two new Hanabi datasets (HanabiLogs and HanabiRewards) for instruction tuning and reward optimization.

**Strengths:**

1. This paper is well-written and well-organized.
2. The paper provides a comprehensive benchmark, which includes several structured context designs, and the dataset could support future studies.

**Weaknesses:**

1. The analysis in this paper remains superficial, just showing the scores without the in-depth analysis.
2. The entire work is constrained to a single game domain and does not show how this benchmark generalizes to the real-world setting.
3. No mention of sampling temperature, top-k/p values or seed control.
4. It is unclear whether models play as independent agents or as a shared model controlling all players sequentially.

**Questions:**

1. How do you operationally define “cooperative reasoning” in this context? What measurable behavior distinguishes it from simple rule-following or pattern matching?
2. How do you justify general claims about “multi-agent reasoning” from a single domain (Hanabi)? Have you tested transfer to any other cooperative tasks?

---

> ### Author Response · Authors · 2025-11-25
>
> We thank the reviewer for their feedback and are glad they found our work **well-organized** and **comprehensive**, with **potential impact on future works**. Below, we address review comments:
>
> ### Weaknesses
> ---
>
> W1. We respectfully disagree that our analysis is limited to scores. In our original submission, we qualitatively discuss emergent strategies for different model families (Sec. 4), analysis of how reasoning models degrade when required to implicitly track state over multiple turns rather than relying on engine deductions (Sec 5), and how team performance is affected by cross-play between strong and weak agents (Sec 6.1). We also analyze failure modes with respect to richness of context (App C), ablations on sampling and multi-agent scaffolding (App E), comparison to human performance (App F), and qualitative analysis of finetuning on our HanabiLogs dataset (App H). In the revision, we also analyze Mycroft with an LLM judge to assess how well agents maintain and update state across turns with new metrics for deduction accuracy, history integration, and state tracking quality (Table 3, App E.3).
>
> W2. We agree that generalization beyond a single game domain is an important problem, but **strong benchmarks studying general cooperative capabilities of LLMs in the context of human strategy** (tracking belief states similar to how humans operate) **do not currently exist to our knowledge**. This is why we created our Hanabi benchmark and evaluation suite. This is especially relevant today, where we expect models to work on extremely long horizons. We also show that training on our dataset improves state-tracking and temporal reasoning. To evaluate the generalizability of our model, in the revision (Section 6.3), we evaluate Qwen3‑4B trained on our HanabiRewards dataset (Sec. 6.2) on downstream tasks and show:
> 1. **Cooperative transfer extends beyond Hanabi** on the recently released Group Guessing Game [1]
> 2. **No degradation or improved performance on downstream tasks**, i.e. long context temporal reasoning (EventQA [2]), general instruction following (IFBench[3]), and mathematical reasoning (AIME2025 [4]).
>
> W3. We thank the reviewer for pointing out missing details, but note that we already share seeding details in App G. We have added further details about temperature, top-k values, and reasoning effort and moved this to App B in the revision.
>
> W4. We provide this detail in Section 3 in our original submission, i.e. “all games are played with each player using the same LLM as a Hanabi playing agent, e.g. four GPT-4.1 agents playing as a four-player team.” We have added a point in the revision to stress that there is no shared model backbone across agents.
>
> ### Questions
> ---
>
> Q1. As we discuss in Appendix A in the original submission, Hanabi is a cooperative, partially observable game where all players share a single team reward, see only their teammates’ cards, and must act through limited, strategically chosen hints. Success requires reasoning about what others know, how they will interpret hints, and how each action advances the shared team objective. In this setting, we use “cooperative reasoning” in a behavioral, operational sense:
> An agent exhibits cooperative reasoning when it (i) uses teammates’ past actions and hints to maintain and update a belief state over hidden cards and teammates’ knowledge, and (ii) selects plays, hints, and discards that improve the joint team score, beyond what can be captured by fixed local rules or static pattern matching. We aim to measure how each agent acts based on its view and prediction about other agents' future actions (Theory of Mind).
> We have added further discussion on this definition in App A.
>
> *``How is it different from simple rule following or pattern matching?’’*
>
> We discuss this point in Appendix A in our original submission. Different player counts require agent adaptation, and 2-player strategies are not equivalent to 2+ player ones, which is also shown by Hu et al. [5]. Moreover, as we detail in Sec 6.1, agents need to adapt to the strategies of other players. We show that unlike traditional RL agents [5], LLMs perform well even in such cross-play settings.
>
> Q2. In our original submission, we evaluate multi-agent cooperative reasoning only on Hanabi via impact of player count (Fig 9, Appendix) richness of context (Sec 5), and cross-play between heterogeneous agents (Sec. 6.1). In the revision, as mentioned above in W2, we also evaluate our HanabiRewards trained model and show it extends to cooperative games beyond Hanabi [1], as well as general purpose tasks (temporal reasoning [2], instruction following [3], and mathematical reasoning [4]).
>
> We hope that the rebuttal clarifies the questions raised by the reviewer. We would be **very happy to discuss any further questions about the work**, and would really appreciate an appropriate increase in the score if the reviewer’s concerns are adequately addressed to facilitate acceptance of the paper.

---

### Author Response · Authors · 2025-11-25
**Rebuttal Summary**

We thank all reviewers for their constructive feedback, questions, and suggestions, all of which helped to clarify and strengthen the contributions of our paper.

The reviewers recognized several key strengths of our work:
1. **Comprehensive and Well-Structured Evaluation (EE5A, PfUB, aPy3, nK8E)**: reviewers found our work clearly written and presenting a broad, systematic evaluation across 17 models, player counts, and context richness.
2. **Benchmark Contributions and Future Impact (EE5A, PfUB, aPy3, nK8E)**: our new datasets HanabiLogs and HanabiRewards were viewed as valuable, forward-looking resources that can support future research in instruction tuning, reinforcement learning, and multi-agent reasoning with Hanabi itself being a meaningful benchmark for advancing research in cooperative multi-agent cooperation.
3. **Transparency and Reproducibility (PfUB, aPy3, nK8E)**: Reviewers appreciated our effort towards trustworthy and replicable results via transparent reporting of metrics, hyperparameters, prompts, and evaluation details.

Given all the reviewer feedback, we summarize our changes in the revision here:

1. **Generalization to downstream tasks (EE5A, nK8E)**: we evaluate Qwen3-4B-Instruct-2507 fine-tuned on our proposed HanabiRewards dataset and show in-domain robustness of cooperative reasoning outside of Hanabi (Group Guessing Game [1]) as well as no degradation or improved performance on general purpose downstream tasks, i.e. long context temporal reasoning (EventQA [2]), general instruction following (IFBench[3]), and mathematical reasoning (AIME2025 [4]).

2. **Statistical Significance of Finetuning on our Datasets**: In our original submission, reviewers raised concerns that the performance gain of Qwen3-4B that we instruction-tuned on HanabiLogs may not be statistically significant [PfUB, nK8E]. To clarify this point, in the revised version, we trained Qwen3-4B-Instruct-2507 with reinforcement learning using our HanabiRewards dataset as verifiable rewards, and **show significant improvement over the base model** in both Sherlock (+6.6 / 25) and Mycroft (+7.5 / 25 i.e. >150%)  setups, **closing the gap to a large proprietary model**, o4-mini, to within 3 points (Fig 7, Fig 8) with a small 4B model. Our finetuned model Qwen3-4B-HanabiRewards outperforms the best non-reasoning model, GPT 4.1, by 4.2 points.

We have addressed the reviewers' questions in detail in our individual responses. We note that based on our revision, reviewer aPy3 highlights that our work is a *"good paper that warrants acceptance"*, and reviewer nK8E highlighted that we *"addressed their primary concern regarding generalisability"*, after which they raised their score to 8 (accept).

### References
---
[1] Riedl, Christoph. Emergent coordination in multi-agent language models. https://arxiv.org/abs/2510.05174

[2] Hu et al. Evaluating Memory in LLM Agents via Incremental Multi-Turn Interactions. https://arxiv.org/abs/2507.05257

[3] Pyatkin et al. Generalizing verifiable instruction following. https://arxiv.org/abs/2507.02833

[4] Aime 2025 benchmark. https://www.maa.org/math-competitions/american-invitational-mathematics-examination.

[5] Hu et al. "Other-Play" for Zero-Shot Coordination. https://arxiv.org/abs/2003.02979

---

### Meta-Review · Area_Chair_EXR2 · 2026-01-06

**Summary:**

This paper presents a large-scale benchmark study of 17 LLMs as cooperative agents in the imperfect-information game Hanabi, evaluating performance across 2–5 players under three increasingly demanding context/scaffolding regimes (Watson/MinCon, Sherlock/DeductCon with engine deductions, and Mycroft requiring implicit multi-turn state tracking). Reviewers questioned statistical significance, generalization beyond Hanabi, and clarity of experimental details.

**Reviewer Concerns:**

1. The submission was criticized for insufficient depth of behavioral diagnosis / unclear operational definition of “cooperative reasoning,” with requests to go beyond raw scores to explain why models succeed/fail (reviewers EE5A, aPy3).
2. Reviewers questioned external validity/generalization beyond a single game (Hanabi) and whether the claims extend to broader cooperative reasoning (reviewers EE5A, nK8E).
3. Some conclusions were questioned due to high variance / missing statistical significance analysis, making it unclear whether observed gains (especially from finetuning) are reliable (reviewers PfUB, nK8E).
4. The paper needed clearer experimental reproducibility details (sampling params, random seeds, whether agents are independent vs. a single controller, prompt availability, and reward/utility generation details) (reviewers EE5A, aPy3, PfUB).
5. Novelty was questioned relative to prior Hanabi/LLM benchmark work, with requests to more clearly articulate what is new beyond existing arenas/benchmarks (reviewer nK8E).

**Reviewer Scores:**

The reviewer may increase their score in light of the authors’ substantial new empirical results; however, the AC does not believe any such update would be sufficient to shift the overall recommendation from reject to accept.

---

### Decision · Program_Chairs · 2026-01-26

Reject